# Dietary Restriction against Parkinson’s Disease: What We Know So Far

**DOI:** 10.3390/nu14194108

**Published:** 2022-10-03

**Authors:** Zhonglei Wang, Yueran Cui, Lulu Wen, Haiyang Yu, Juan Feng, Wei Yuan, Xin He

**Affiliations:** 1Department of Neurology, Shengjing Hospital of China Medical University, Shenyang 110004, China; 2Department of Orthopedics, The First Hospital of China Medical University, Shenyang 110001, China

**Keywords:** dietary restriction, Parkinson’s disease, gut microbiome, mechanism, patients, treatment, prevention

## Abstract

Dietary restriction (DR) is defined as a moderate reduction in food intake while avoiding malnutrition. The beneficial effects of DR are being increasingly acknowledged in aging and in a series of age-related neurodegenerative disorders, for example, Parkinson’s disease (PD). To date, the pathogenesis of PD remains elusive and there is no cure for it in spite of intensive research over decades. In this review, we summarize the current knowledge on the efficacy of DR on PD, focusing on the underlying mechanisms involving general metabolism, neuroendocrinolgy, neuroinflammation, gut microbiome, and so on. We anticipate that this review will provide future perspectives for PD prevention and treatment.

## 1. Introduction

Parkinson’s disease (PD) is the second most common neurodegenerative disease, with an incidence of 15/100,000/year and prevalence of 100–200/100,000 [1,2], imposing a growing socioeconomic burden globally. To date, PD is still incurable and its pathogenesis remains elusive [3,4]. Evidence from experimental studies reveals that mechanisms including protein misfolding and aggregation, neuroinflammation, mitochondrial dysfunction, and altered gut bacteria composition contribute to PD development [2,5,6,7,8,9]. In addition, PD is extremely heterogeneous regarding the age of onset, clinical manifestations, rate of progression, and therapeutic responsiveness [10,11]. As of now, a number of medication strategies have been widely applied to control the motor and non-motor symptoms of PD and improve the quality of life, among which levodopa remains the most effective [4,12,13,14]. Other common medications include dopamine agonists [15,16,17], monoamine oxidase B inhibitors [18,19,20], amantadine, and anticholinergic medication [4,12]. However, with long-term application of these drugs and as the disease progresses, adverse effects like gastrointestinal reactions, motor complications, psychotic disorders, and decreased efficacy emerge [12,13,14]. Moreover, these medications could neither effectively prevent the disease onset nor stop the disease progression. Recently, lifestyle interventions in the promotion of healthy brain aging and the prevention and treatment of central nervous system (CNS) diseases have risen into the spotlight [21,22], which could be promisingly complementary to the conventional PD pharmacotherapy.

Dietary restriction (DR), which involves a moderate reduction in food intake while avoiding malnutrition, has been proven to be effective in holding back aging and relieving age-related chronic diseases, including cancer, cardiovascular disease, diabetes, and neurodegenerative disorders [23]. The beneficial actions of DR involve metabolic, hormonal, and immunomodulatory mechanisms. DR could reduce obesity and visceral fat, thus preventing metabolic risk factors [24]. It increases insulin sensitivity, glucose tolerance, and ghrelin level [25,26]. It can also induce adipose tissue transcriptional reprogramming, involving ways to regulate mitochondrial bioenergy, anti-inflammatory response, and longevity [27]. Moreover, a potential role of DR in regulating the gut–brain axis has been well described in diseases of CNS and intestinal microbiota transplantation has been shown to be effective in Alzheimer’s disease (AD) and multiple sclerosis (MS) [28,29]. Here, we summarize the strategies of DR from clinical and laboratory studies, and review the current findings of DR in preventing and ameliorating PD, with an emphasis on the possible mechanisms.

## 2. Dietary Restriction

### 2.1. Methods of DR

DR strategies include long-term reductions in food intake or changes in the frequency and interval between meals [22,30,31]. Methods of DR vary in animal and clinical studies [32,33]; here, we describe the common dietary intervention strategies.

#### 2.1.1. Calorie Restriction

Calorie restriction (CR) refers to a 10–50% reduction in daily food intake without changing the frequency of eating [22,34,35]. In humans, a two-year 25% CR has been reported to improve overall health in non-obese participants with few adverse effects [36]. In caenorhabditis elegans, the gene Sir-2.1 that mediates longevity [37,38] is widely expressed under CR induction [39].

#### 2.1.2. Intermittent Fasting

Intermittent fasting (IF) refers to eating little or no food for a longer period of time (e.g., 16–48 h) and eating normally for the rest of the time [40]. In different studies, the specific protocols for achieving IF also vary.

(a)Alternate day fasting (ADF)

In laboratory rodents, alternate day fasting is the most commonly used type of IF besides time-restricted feeding (TRF) [40]. It is also known as every-other-day fasting (EODF), which achieves a comparable effect to CR on reducing fasting blood glucose and insulin concentrations [41]. This dietary strategy has been confirmed by clinical trials for its safety and health promotion effects like weight loss and reducing the risk of cardiovascular diseases [41,42].

(b)Time-restricted feeding

TRF is achieved by restricting the time window of food intake, with normal eating within it [34,43]. Interestingly, the mechanism of TRF to prolong lifespan and attenuate age-related cardiac decline is not equivalent to that of CR, which may be accomplished by enhancing the expression of circadian rhythm genes. The combined effect of the molecular circadian clock and the imposed feeding and fasting rhythms can improve the rhythm of gene expression under TRF and provide systemic metabolic benefits [43,44].

(c)Periodic fasting

Periodic fasting refers to a type of arrhythmic IF that often involves 5 days without energy restriction and 2 days with less than 75% energy (intermittent calorie restriction (ICR)) [34,45]. Such a strategy has been proven to protect against diabetes, cancer, heart disease, and neurodegeneration in rodents; in humans, it is effective in reducing obesity, hypertension, asthma, and rheumatoid arthritis [46,47].

#### 2.1.3. Fasting Mimicking Diet (FMD)

FMD is also known as periodic fasting or cycles of fasting, which is based on severe calorie restriction of 50% or more, low protein and sugar, with a relatively high fat content. It has been suggested to prolong lifespan and prevent cardiometabolic risk [48], as well as promote neural myelination, suppress autoimmunity [49], and induce cancer regression [50].

### 2.2. The Physiological Benefits of DR

From yeast and fruit flies to monkeys and humans, DR has shown positive effects of extending lifespan, promoting metabolism, preventing chronic diseases and neurodegeneration, and improving overall health [31,47,51]. Based on yeast models, researchers have revealed that the lifespan-extension effect of DR is related to mechanisms including preserving mitochondrial respiration [52], promoting Snf1 activation [53], regulating cell cycle [54], and so on. Moreover, DR could notably improve glucose metabolism and promote mobilization of fatty acids [55]. In addition, IF has been demonstrated to reduce blood pressure, increase heart rate variability, and reduce insulin resistance, as reviewed elsewhere by Dr. Mattson [40]. In AD transgenic mice, DR can alleviate cognitive deficits, amyloid lesions, and microglial reactivity [56]. In addition, prophylactic IF can reduce tissue damage and neurological deficits after ischemic stroke by inhibiting excitotoxicity, oxidative stress, and inflammation [57]. Moreover, DR also exerts positive effects on healthy obese people; after a month of 15–16 h daily fasting, the body weight, BMI, and fat content in healthy obese people were significantly reduced, accompanied with notably elevated gastrointestinal hormones that regulate satiety, including leptin, glucagon-like peptide-1 (GLP-1), peptide tyrosine-tyrosine (PYY), and cholecystokinin (CCK), except for ghrelin. It is suggested that IF can be used as a strategy against obesity in the healthy population [58]. Analysis of data based on CALERIE phase 2 (CALERIE 2), the largest cohort study to date of sustained CR in non-obese healthy humans [59], showed that a two-year CR could produce parallel beneficial effects as in the animals receiving a similar intervention [60]. Besides improving mood, sleep, quality of life, and overall health [36], it also reduces inflammation and may prolong the health span by inhibiting the Pla2g7 gene [27], which encodes platelet activating factor acetyl hydrolase (PLA2G7), as well as inflammatory activators associated with a variety of diseases like cancer, neuropathy, and stroke [61]. Moreover, DR has been shown to exert immunomodulation by regulating multiple signaling pathways, for example, promoting the targeted regulation of rapamycin (mTOR) in mammals and the activation of AMP-activated protein kinase (AMPK) [62,63,64]. Inactivation of mTOR induces a metabolic transition from glycolysis to oxidative phosphorylation and counteracts the chronic activation of glycolysis in senescent T cells [65]. Activated AMPK improves immune function by promoting mitochondrial regeneration through the expression of nuclear respiratory factor (NRF)-1 and peroxisome proliferator-activated receptor (PPAR) g transcription factor [66,67]. However, it should be noted that chronic CR is not fully equivalent to intermittent energy restriction and may instead impair certain immune functions by inhibiting mTOR [67]; further, optimal dietary composition and fasting cycles remain to be defined.

### 2.3. Life-Extending Effect of DR and the Influencing Factors

DR can extend the lifespan of yeast by threefold, the lifespan of worms by two- to threefold, the lifespan of flies by twofold, and the lifespan of mice by 30–50% [51]. Despite the potential advantages of lifelong DR to slow aging and lengthen life span, it is hard to carry it out for a life-long period. DR started late in life may be more practical [68], but can dietary interventions that start at an older age have similar effects to those that start younger? DR started at 6 months of age did not increase the lifespan, but a reduction in mean lifespan of 15% was observed in A/J mice of 10 months old [69]. In another study, male C57BL/6 mice were fed either 40% DR or an ad libitum (AL) feeding scheme from 3 to 12 months of age and then were either switched to the other feeding regimen or kept in the same state for an extra 3 months. Both 9 and 12 months of DR regimens (long term), along with 3 months (short term, mid-life beginning) of DR, alleviated DNA damage and reduced the adipocyte size in the visceral fatty tissue with similar effectiveness, suggesting a metabolic retention of the DR phase [70].

In addition to the starting age of DR, the influence of gender should not be underestimated [71,72,73,74]. A 40% DR could significantly improve glucose tolerance [75,76], which has been proposed to play a role in the life-extending effects of DR in the male mice, and such a metabolic change could be maintained for two months after switching to a free diet [77], while in the female mice, such a transition not only resulted in a rapid loss of RNA expression patterns associated with DR, suggesting no or only weak memory of the previous DR regimen, but also significantly increased mortality in these mice after transition [78]. Moreover, the time spent carrying out DR is equally important [44,47,79]. In one study of intermittent TRF (iTRF), nighttime-biased fasting could extend the lifespan of Drosophila flies and enable them to show better flight ability and cardiac and flight muscle function, but fasting during the day, as opposed to what the circadian clock dictates, could abolish the benefits of iTRF. Long et al. suggested that such a phenomenon could be explained by enhanced autophagy at night [44]. In conclusion, the effect of DR on lifespan is affected by multiple factors, such as age, gender, and genotype, and the mechanism remains to be further explored.

### 2.4. Delay Aging and Age-Related Diseases

DR delays aging [80] and reduces the risk of age-related diseases like diabetes, cancer, cardiovascular disease, and neurodegenerative diseases in mounting studies based on laboratory animals and humans [42,51,81,82]. Aging is a key risk factor for neurodegenerative diseases [83,84,85] and DR can improve the adverse effects of aging on the brain [86,87]. Rumani Singh et al. found that 24-month-old Wistar strain male rats (equivalent to completing 70% of their lifespan) experiencing a 3-month ADF program showed improved age-related motor coordination, as well as learning and memory function, compared with the free-fed rats, despite that all of the indicators were inferior to the 3-month-old young control group. Meanwhile, fasting significantly reduced the protein carbonyl content in the cerebral cortex, hippocampus, and hypothalamus, which increased remarkably with aging, and it partially restored the decreased Syn and CaM expression in the hippocampal CA3 and DG areas, piriform cortex, and hypothalamus [88]. Moreover, DR could ameliorate late-life depression, which is increasingly acknowledged in elder people and is a key non-motor symptom of PD [89,90,91,92]. A community project for 90-year-olds in Italy showed a significant positive correlation between body fat and depression in older adults [93] and that CR can work as an antidepressant [94]. In a randomized controlled experiment, Rebecca et al. reported that, in overweight and obese women with polycystic ovary syndrome, DR improved depression assessed with CES-D scores [95]. Further, in rodent models, CR for just 10 days led to significant antidepressant responses [96]. In addition, community survey evaluations showed that myeloperoxidase levels were positively correlated with mortality in older individuals [97], while CR is able to attenuate the increased myeloperoxidase activity during aging [98].

Metabolically, DR can improve hyperglycemia, glucose tolerance, and insulin sensitivity in AL-fed aging mutant and normal mice, and reduce the consequences of aging-induced glycemic dysregulation, such as hepatic gluconeogenesis [99]. A recent randomized controlled trial in healthy middle-aged adults demonstrated that, after a 4-week ADF diet regimen, not only did cardiovascular indicators improve, but on fasting days, the pro-aging amino acid methionine was periodically depleted; additionally, after long-term ADF, levels of sICAM-1 (an age-related inflammatory marker), low-density lipoprotein, and the metabolic regulator triiodothyronine were all reduced. Such a fasting regimen showed no adverse effect even after six months [42], laying the basis for the safety and feasibility of ADF as an effective clinical intervention against aging and age-related disorders.

## 3. General Neuroprotective Effects of DR

### 3.1. Improve Cognitive and Motor Function

Aging causes changes in the structure, physiology, and metabolism of the brain, eventually leading to impaired cognitive and motor-neural abilities [100,101,102]. Fasting exerts positive effects on the brain via metabolic, cellular, and rhythmic mechanisms, directly or indirectly, which could enhance cognitive performance and protect against CNS disorders [100,103]. It is reported that 18-month-old rats on a 60% CR diet presented a higher survival rate, better locomotor activity, and a delayed age-related decline in cognition [104]. In addition, monkeys treated with MPTP that underwent 30% CR could accomplish more specified work compared with those with a normal diet, despite there being no statistical difference [105]. Another study on rats at the age of 24 months also demonstrated that even a late-onset short-term IF-DR regimen (EODF) could well relieve the age-associated decline in motor coordination and cognition performance [88]. Another clinical trial in 43 individuals with central obesity aged 35–75 years revealed that a four-week dietary intervention (either CR or IF) could improve cognition, but the intermittent energy restriction group showed a notable deterioration in recognition memory [106]. Fu et al. reported that different forms of CR exerted diverting effects on spatial recognition memory in developing male mice; that is, chronic CR showed no effect on the preference of mice for novelty in the Y-maze, but severe CR could impair spatial recognition memory, while acute CR could either improve or impair spatial recognition memory depending on the individual [107]. According to a randomized clinical study in 48 subjects under 6 months of CR, the energy deficit was not related to a consistent pattern of cognitive deficits [108]. These findings indicate that DR could affect cognitive and motor function differently depending on the intensity, duration, and time of initiation. However, it is plausible that mild to moderate DR started early in life is more likely to yield beneficial effects on the brain.

### 3.2. Promote Neurotrophic Factor Levels

Neurotrophic factors are critical mediators of neuronal plasticity by promoting axonal and dendritic development and remodeling [109,110]. They are involved in the generation of neurotransmitters as well as the formation of synapses [111]. A 6-month 30% CR regimen in monkeys has been proven to lead to three times higher glial-cell-line-derived neurotrophic factor (GDNF) levels in the right and left caudate nucleus (CN) compared with monkeys fed ad libitum [105]. Moreover, the DR regimen could increase the level of brain-derived neurotrophic factor (BDNF) dramatically in the hippocampus, cerebral cortex, and striatum, and the subsequent activation of BDNF signaling pathways plays a critical role in the neuroprotective effect of DR [112,113]. BDNF regulates the baseline level of neurogenesis in the dentate gyrus of adult mice and augments neurogenesis by DR [113]. Furthermore, formalized paraphrase in db/db mice, a model of insulin-resistant diabetes, CR, wheel running, or a combination of both increased the hippocampal BDNF level, which was accompanied by increases in dendritic spine density on the secondary and tertiary dendrites of dentate granule neurons. Moreover, in post-ischemic mouse models, a hypocaloric diet for 8 weeks could well confer neuroprotection and promote peri-infarct brain remodeling by upregulating BDNF [114].

### 3.3. Improve Neuronal Plasticity

Neuronal plasticity is a gradual process of the formation of neuronal structure and function during development and through learning. It involves trophic processes like neurogenesis and synaptogenesis as well as atrophic processes like degeneration of inactive neurons and neuronal contacts [115,116]. Despite there being much to be explored, there is mounting evidence that DR modulates brain structure and function throughout the lifespan [88,117]. In aging male rats, the expression of synaptic plasticity-related proteins, such as synaptophysin, calcineurin, and CaM kinase II, in the hippocampal CA3 and DG regions, piriform cortex, and hypothalamus are significantly reduced compared with young mice, but could be partially recovered after 3 months of ADF [88]. In addition, ADF significantly improved brain function and structure, as demonstrated by better learning and memory assessed with the Barnes maze and fear conditioning, a thicker CA1 pyramidal cell layer, greater expression of the dendritic protein drebrin, and lower oxidative stress, in the IF mice compared with mice fed ad libitum [118]. Besides, IF has been reported to effectively promote the regeneration of artificially crushed mouse sciatic nerve axons by promoting indole-3-propionic acid (IPA) production by the intestinal microbe Clostridium sporogenes, which can significantly improve epidermal innervation, thereby promoting sensory recovery [119]. A detailed description of the physiological and neuroprotective effect of DR is listed in Table 1.

## 4. Effects of DR on PD and the Underlying Mechanisms

The key pathological features of PD include the formation of Lewy body [140] and the loss of dopaminergic neurons [2]. It is now established that genetic factors, α-synuclein misfolding and aggregation [141], neuroinflammation [5,142,143], oxidative stress [144,145], mitochondrial abnormalities [6], impaired protein clearance [146,147,148], and intestinal flora disorders and intestinal inflammation [149,150] all contribute to the onset and progression of PD.

### 4.1. DR and PD Animal Models

Despite that the pathogenesis of PD remains to be explored, several PD animal models have been well established: (1) The 1-methyl-4-phenyl-1,2,3,6-tetrahydropyridine (MPTP) mouse model is the most commonly used PD animal model [151]. The subacute model is widely chosen because it could recapitulate PD progression, with the drawback of failing to elicit notable motor deficit. However, it could disrupt the blood–brain barrier (BBB) in the substantia nigra (SN) pars compacta (SNpc), activate astrocytes and microglial NLRP3 inflammasome in the striatum, upregulate α-synuclein expression, and induce Lewy-body-like pathology [151,152]. (2) The neurotoxin 6-hydroxydopamine (6-OHDA) intracerebral infusion is a classic PD model that causes massive destruction of nigrostriatal dopaminergic neurons [153]. Unlike the MPTP model, it is mainly used for the study of motor and biochemical dysfunction in PD. Although its manipulation carries a lower risk of toxicity, the effects of 6-OHDA in the brain appear to be heterogeneous and strongly influenced by the site of infusion [154]. (3) It is speculated that the α-synuclein pathology of PD originates in the gastrointestinal tract and spreads to the brain through the vagus nerve [155]; however, previous classical PD models fail to fully copy such a pathological process. It has recently been reported that, after injection of α-synuclein preformed fibers (PFFs) into the muscularis of the mouse duodenum and pylorus, pathological α-synuclein aggregates could be detected in the dorsal motor nucleus of the vagus (DMV) and locus coeruleus, which subsequently spread to the amygdala, dorsal raphe nucleus, and SN [156,157]. This model provides the most accurate characterization of α-synuclein gut-to-brain transmission to date and makes it possible to prevent PD by interfering with the production of pathological α-synuclein in the gut [158,159] (Figure 1). (4) A53T SNCA transgenic mice displayed a very similar expression pattern of human α-synuclein to endogenous mouse α-synuclein. They express truncated, oligomeric, and proteinase K-resistant phosphorylated forms of α-synuclein in areas specifically affected by PD and/or dementia with Lewy bodies, including the olfactory bulb, cerebral cortex, striatum, and SN [160].

With these animal models, the effects and the potential mechanisms of DR on PD have been increasingly studied [105,136,161]. In the MPTP-induced PD model, DR could decrease the neurotoxicity of MPTP on dopaminergic neurons by inducing stress proteins like heat-shock protein 70 and glucose regulate protein 78 [159]. Moreover, fasting for 3 days followed by feeding for 4 days, an FMD scheme, preserved motor function and attenuated MPTP-induced loss of SN dopaminergic neurons. BDNF, which promotes dopaminergic neuron survival, is elevated in PD mice after FMD, suggesting that BDNF is involved in FMD-mediated neuroprotection [132]. Additionally, adult male rhesus monkeys maintaining a 30% CR for 6 months and injected with MPTP into the right carotid artery showed significantly higher motor activity and higher levels of dopamine and dopamine metabolites in the striatal region compared with AL controls. Moreover, a notable increase in GDNF was also observed, suggesting that more than one form of neurotrophic factors mediate the anti-PD effect of CR [105]. In the yeast model, CR is performed by reducing the glucose concentration in the initial medium from 2% to 0.5% or 0.05%. Such a regimen yielded prolonged longevity and reduced α-synuclein-mediated toxicity, which were mediated by the maintenance of autophagy steady state [162]. SNCA mice expressing A53T mutant human α-synuclein under the control of the mouse Thy-1 promoter that causes familial PD have impaired autonomic control of their heart rate, which is exacerbated by a high-energy diet via a mechanism involving increased susceptibility of parasympathetic brainstem neurons to α-synuclein, and this could be partially restored by ADF [163,164]. Moreover, in normal rats of non-PD models, there is a considerable and continuous rise in α-synuclein mRNA expression in the cortex and hippocampus of rats with aging, while long-term DR could reverse the late age-related increase in α-synuclein expression [136]. More PD models based on different mechanisms are needed to fully elucidate the specific effect of DR on PD; for instance, whether DR can prevent PD progression in the PFF-induced PD model remains unknown.

### 4.2. DR and PD-Related Risk Factors

Regarding the potential risk factors for PD, a meta-analysis of seven observational cohort studies showed that people with diabetes had a 38% increased risk of developing PD [165]. It is speculated that diabetes may be an important confounding factor in PD pathogenesis [166]. Insulin signaling has been proven to cause neurodegeneration by affecting insulin dysregulation, amyloid aggregation, neuroinflammation, mitochondrial dysfunction, and altered synaptic plasticity [167]. In mice under 12 weeks of a high-fat diet (HFD), glucose intolerance promotes endogenous neurotoxin accumulation, which damages dopaminergic neurons in the brain and is associated with PD-like slowness and instability of gait [168]. Another well-studied risk factor of PD is obesity, which is associated with multiple pathological pathways, including metabolic abnormalities, altered hormonal signaling, and increased inflammation, which contribute to downstream neuropathology [169]. Gene-Jack Wang et al. reported that striatal dopamine D2 receptor availability is significantly lower in obese individuals than in the controls [170]. As a weight loss strategy, IF cannot only reduce body weight in the patients with type 2 diabetes (T2DM), but also lower glycated hemoglobin [171]. In db/db mice, an insulin-resistant diabetes model, the reduction in BDNF levels in the hippocampus and the density of dendritic spines on secondary and tertiary dendrites of DG granule neurons could be partially restored by CR, and in wild-type mice, CR showed an additive effect on the elevation of BDNF levels in the hippocampus [172]. Recently, Curie Kim et al. studied the effect of intermittent energy restriction (IER) and continuous energy restriction (CER) on cognition related to neurogenesis in the human hippocampus, and they found that both CER and IER improved adult-hippocampal neurogenesis-dependent cognition in adults with central obesity [106]. Moreover, increased β-cell death and impaired autophagic flux are present in the islets of obesity-induced diabetic mice, while IF could restore the islet autophagic flux and enhanced β-cell survival through the autophago-lysosomal pathway. Meanwhile, IF could also improve glucose tolerance and lower fasting blood glucose in these diabetic mice by enhancing glucose-stimulated insulin secretion and nuclear expression of the pancreatic regeneration marker NEUROG3 [129]. So far, no clinical guidelines have been published regarding the implementation of dietary interventions like IF in patients with diabetes [173], and more clinical data are needed to assess the safety and efficacy of such dietary regimens. As another risk factor for PD, the relationship between traumatic brain injury and neurodegenerative diseases is complex [174], but both IF and CR after traumatic brain injury have been proved to be neuroprotective by alleviating mitochondrial dysfunction, promoting hippocampal neurogenesis, inhibiting glial cell response, shaping nerve cell plasticity, and modulating autophagy and apoptosis [175,176].

### 4.3. Possible Mechanisms of DR on PD

#### 4.3.1. Ameliorate Neuroinflammation

Neuroinflammation is the coordinated response of the central nervous system to noxious stimuli, for example, infection, traumatic brain injury, or other neurological disorders, and chronic neuroinflammation may take place if the predisposing factors persist [22,177,178]. Postmortem brain pathology of PD patients revealed inflammatory changes in microglia and elevated levels of proinflammatory mediators like IL1β, IL6, and TNFα in the striatum [5]. Microglial NLRP3 inflammasome activation induces dopaminergic neuron death and subsequent dyskinesia in PD, while dopamine can significantly reduce NLRP3-dependent inflammasome activation in mixed glial cells stimulated by NLRP3 agonists. It is indicated that the reduction in dopamine levels in PD may exacerbate neuroinflammation in the CNS by hindering the ability of dopamine to inhibit the inflammasome, thus forming a vicious circle [152].

Meanwhile, the NLRP3 inflammasome can assemble under stimulation by accumulated endogenous metabolites, such as fibrillar amyloid β and 25-hydroxycholesterol, leading to the activation of caspase-1 and subsequent secretion of IL-1β [152]. The former can truncate the normal multimeric α-synuclein into a form that is prone to aggregation, eventually leading to neuronal dysfunction and loss [179]. Chronic systemic expression of the latter is also able to exacerbate neurodegeneration and microglial activation in the SN. Furthermore, the downstream molecule NO of IL-1β action is partly responsible for the worsening of neurodegeneration observed in PD [180]. In mice, prolonged (48 h) fasting blunted NLRP3 inflammasome activation and reduced IL-1β levels [181]. This is consistent with previous observations in humans [182]. Mechanistically, the negative regulation of fasting-induced SIRT3 on NLRP3 is achieved through SIRT3-mediated mitochondrial SOD2 deacetylation, resulting in SOD2 activation. Therefore, the anti-inflammatory effects of the mimic fasting diet may be mediated in part through SIRT3-directed blunting of NLRP3 inflammasome assembly and activation [181]. Recently, peripheral circulating CD4+ and CD8+ T cells from PD patients were shown to produce cytokines in response to α-synuclein. Further, α-synuclein overexpression in the mouse midbrain results in upregulation of major histocompatibility complex II (MHCII) protein in CNS myeloid cells along with T cell infiltrating into the CNS; such an enhanced T-cell response primarily involves increased CD4+ T cells that produce the cytokine IFNγ, while knockout of CD4 T cells could attenuate α-synuclein induced neurodegeneration [183]. Francesca Cignarella et al. found that the proportion of CD4+ T cells in the draining lymph nodes of the IF group as well as the level of IL-17A and IFN-γ were lower than those of the free diet group in the multiple sclerosis (MS) mice, and thereby ameliorated symptoms in the EAE mice. In addition, IF improved T cell composition in the mouse gut lamina propria, thereby modulating systemic immunity [133]. Considering the gastrointestinal inflammation in PD [149], whether IF can play a similar role in PD is worth exploring.

#### 4.3.2. Reduce Oxidative Stress

Reactive oxygen species (ROS) are the products of normal cellular metabolism and mediate a wide range of physiological processes, but at high concentrations, they can adversely modify biomolecules like proteins, lipids, and DNA [184,185]. When the antioxidant defense system in the cell fails to balance oxidants, the ratio of oxidants to antioxidants is disturbed and the oxidants dominate, thus forming oxidative stress [186]. Consequently, the defense mechanisms are unable to cope with ROS-induced damage [187]. DA metabolism, high levels of iron and calcium in SN, mitochondrial dysfunction, and neuroinflammation lead to increased oxidative stress and loss of dopaminergic neurons in the brains of PD patients [188,189]. While many antioxidants, such as creatine, vitamin E, coenzyme Q10, pioglitazone, melatonin, and deferoxamine, have been tested in clinical trials as potential PD treatments, none have been shown to improve neurodegeneration in PD patients [188,190]. Besides the dietary supplements, many studies have demonstrated the remarkable effect of DR in ameliorating oxidative stress. 

Sirtuins (SIRTs 1–7) belong to the nicotinamide adenine dinucleotide (NAD)-dependent histone deacetylases family and play important roles in the regulation of oxidative stress, of which SIRT1, SIRT3, and SIRT5 protect cells from activity oxygen damage and SIRT2, SIRT6, and SIRT7 regulate key oxidative stress genes and mechanisms [191]. There is increasing evidence showing that modulation of sirtuins’ activity by DR regimens can prevent neuronal loss and brain damage in PD models [192,193]. mTOR activation mediates neuronal apoptosis under oxidative stress; for example, the environmental toxin cadmium-induced ROS activates mTOR signaling, leading to neuronal cell death. Mechanistically, calcium/calmodulin-dependent protein kinase II (CaMKII) and calcium signaling pathways are involved in this process [194]. Meanwhile, IF can suppress mTOR levels [195], induce Sox2 and Ngn3 expression, and promote insulin production in pancreatic islets of human type 1 diabetic patients [196]. However, studies have also shown that mTOR activators can be used to prevent oxidative-stress-related neuronal cell death and protect dopaminergic neurons from oxidative stress in PD [194]. Different signaling pathways of DR on mTOR may be involved and more deeper research based on different PD models is necessary.

#### 4.3.3. Preserve Mitochondrial Function and Reduce Mitochondrial Damage

Post-mortem studies in PD patients have shown lowered reactivity of the complex I of the respiratory chain in the SN as well as mitochondrial morphological and DNA losses [145]. Furthermore, mitochondrial failure has been identified in sporadic and hereditary types of PD, along with toxin-induced models [6]. DR can boost the heart mitochondrial respiratory rate while lowering cardiac ROS generation [197] and has a favorable effect on mitochondria by increasing mitochondrial biogenesis and efficiency [187]. Rumani Singh et al. discovered that the activity of the mitochondrial electron transport chain (ETC) decreased significantly in old (24-month-old) ad libitum (OAL) fed rats compared with their young (3-month-old) counterparts, but it could be reversed partially by the IF-DR regimen [88]. Meanwhile, chronic hypoxia-induced CR induces notable changes in myocardial mitochondrial metabolism and is responsible for better diffusion of ADP toward adenine nucleotide translocase, which may be mediated by enhanced permeability of the outer mitochondrial membrane [198].

#### 4.3.4. Maintain Autophagy Homeostasis

Autophagy is a major, conserved cellular pathway through which cells transport cytoplasmic contents to lysosomes for degradation [199], which helps remove misfolded or aggregated proteins, as well as recycle damaged cellular components [200]. In PD, increased intracellular pathogenic α-synuclein can lead to Rab1a-related mislocalization of the autophagy protein Atg9, which inhibits autophagosome biogenesis [201]. In addition, the accumulation of α-synuclein aggregates can not only inhibit the in vivo maturation of autophagy and fusion with lysosomes, but also impair lysosome function, thereby reducing the ability to clear α-synuclein [199]. On the other hand, neuronal α-synuclein activates microglia, which engulf α-synuclein into autophagosomes for degradation through selective autophagy [202]. Injured microglial autophagy exacerbates pro-inflammatory responses by modulating the NLRP3 inflammasome and eventually exacerbates MPTP-induced neurodegeneration [146]. However, in the later stage of PD, overactive or dysfunctional autophagy can also lead to neuronal cell death [200]. Therefore, regulating autophagy homeostasis in PD is a promising target. 

Dietary interventions like fasting and CR are the most potent non-genetic autophagy regulators that act in a variety of organs and tissues, including nerves, liver, kidney, and heart [200]. Autophagy is maintained at homeostatic levels following CR treatment in yeast cells expressing α-synuclein, and such autophagic homeostasis is able to reverse the decreased activity of the α-syuclein-promoted ubiquitin-proteasome system (UPS), another crucial α-synuclein degradation pathway [203]. In AD mice, Xigui Chen et al. showed that fasting increases the number, size, and signal strength of neuronal autophagosomes in the brain and, for the first time, they discovered a circadian rhythm of autophagy, that is, autophagy parameters increase during the day and decrease during the night [204]. In parallel, Matt Ulgherait et al. applied Drosophila to demonstrate that nighttime-specific fasting significantly prolongs lifespan, while only fasting during the day has no effect on lifespan [205]. In conclusion, DR in a specific time period may have the best effect on autophagy regulation and more research is needed to reveal the effect of DR on selective autophagy in microglia.

#### 4.3.5. Regulate Gut Microbiota Composition and Richness

Recently, an increasing number of studies have highlighted the important role of the gut microbiota in neurological disorders, including AD [28], autism [206,207], MS [29,208], PD [132,149,209,210], and stroke [211,212]. The gut microbiota affects behavior, modulates the production of neurotransmitters in the gut and brain, and influences brain development and myelination patterns. Moreover, the dietary components are chemically transformed by the microbiota so that gut-derived metabolites spread to all organs, eventually affecting the brain [213]. The role of gastrointestinal inflammation in PD can be traced back to the well-known Braak hypothesis [155]: PD may originate in the gut as there may be a pathogen that can pass through the gastric epithelial layer to induce the misfolding and aggregation of α-synuclein in the submucosal plexus and ultimately project to the brain [155,214]. At the same time, multiple surveys have shown that patients with Helicobacter pylori infection and patients with inflammatory bowel disease have an increased risk of PD [215,216]. Recent studies have confirmed the hypothesis that a pro-inflammatory gut environment due to gut hyperpermeability and/or dysbiosi can induce or exacerbate PD, increase α-synuclein levels in the colon, and reduce DA and DA metabolites (DOPAC, HVA) in the striatum [217]. Mechanistically, short-chain fatty acids (SCFAs), one of the metabolites of gut microbes, can regulate gut permeability and blood–brain barrier (BBB) permeability by upregulating tight junction proteins [218]. In PD patients, there is a lack of correlation between SCFA-producing gut bacteria and fecal levels of SCFAs. PD patients show reduced fecal SCFAs, but increased plasma SCFAs, which may be due to increased intestinal permeability, which allows SCFAs to enter the systemic circulation, indicating a dysfunctional intestinal barrier [219]. Both human and animals under a CR or IF diet can develop a physically balanced composition of gut microbiota [133]. Zhou et al. have reported that TRF is effective in preserving motor function and attenuating dopaminergic neuron loss in MPTP-induced PD mice via promoting a favorable gut microbiota composition and metabolites [132]. Clinically, colon cleansing and a dietary intervention incorporating SCFAs have a beneficial impact not only on the gut microbiome, but also on the clinical course of PD [220]. In patients with metabolic syndrome, IF can induce profound changes in the gut microbial community, increase SCFA production, decrease circulating levels of lipopolysaccharides, and significantly reduce cardiovascular risk factors [122]. In addition, IF alters the gut microbiota composition of MPTP-induced PD mice, inhibiting the number of pro-inflammatory bacteria Clostridium in the gut and reducing the number of glial cells as well as the release of inflammatory factors such as IL-1β and TNF -α, thus exerting a neuroprotective effect, despite with no change in the SCFA level [132]. Even though many studies have demonstrated a close correlation between gut microbiota and PD, the mechanism remains unclear. Inflammation, metabolites, or specific probiotics may play a role (Figure 1 and Figure 2).

## 5. Other Dietary Interventions on PD

### 5.1. Low Fat Diet

A high-fat diet (HFD) severely reduces striatal dopamine, SN microtubule-associated protein 2, manganese superoxide dismutase, and tyrosine hydroxylase levels in MPTP subacute mice, and significantly increases mortality of the MPTP acute mice [221]. Multiple studies have shown that a high fat intake can lead to excess production of circulating free fatty acids and systemic inflammation. Immune cells, free fatty acids, and circulating cytokines reach the hypothalamus and initiate local inflammation through several processes such as microglial proliferation [222,223,224,225]. In a randomized controlled trial, PD patients were randomly classified as low-fat diet (LFD) or ketogenic diet for 8 weeks, and both groups showed significantly reduced MDS-UPDRS scores and a notable improvement in both motor daily living experiences and non-motor symptoms like fatigue, sleepiness, and cognitive impairment [226]. In the mice under LFD combined with 40% CR starting at 24 months old, white matter microglia activation was reduced, as indicated by a number of phagocytic markers like Mac-2/Lgals3, Dectin-1/Clec7a, and CD16/CD32, which may explain the protective effects of CR during aging-related decline [227]. In addition, in 6-OHDA-induced PD rats, a high-fat diet induced insulin resistance and significantly increased DA consumption in the SN, while later conversion to LFD reversed such a phenomenon and improved proteins associated with mitochondrial function (e.g., AMPK and PGC-1α) and proteasome function (e.g., TCF11/Nrf1) [228].

### 5.2. Protein-Restricted Diet and Amino-Acid-Restricted Diet

There has been suggested a significant positive correlation between PD mortality and per capita total dietary protein consumption and meat consumption [229]. As early as 1987, low-protein dietary strategies have been clinically developed to reduce fluctuations in PD patients treated with levodopa [230,231,232]. This phenomenon is due to a significant increase in plasma concentrations of large amounts of neutral amino acids (LNAAs) caused by a conventional high-protein diet, which compete with levodopa for transportation across the BBB [233,234,235]. The results of the dietary pattern showed that severe motor fluctuations induced by daytime levodopa or carbidopa treatment were significantly alleviated by eliminating dietary proteins from breakfast and lunch, and the patient’s overall function and sensitivity to the drug improved significantly [236]. There were no adverse effects during the one-year follow-up period [236]. Another recent study also reported that patients with a high-protein diet had younger onset, greater motor fluctuations, and higher rates of PD family members [237]. We can infer that the protein-restricting diet may be more suitable for younger PD patients or familial PD patients. Furthermore, obesity, insulin resistance, and T2DM were associated with elevated levels of branched-chain amino acids (BCAAs) as risk factors for PD. Restricting BCAAs can improve metabolic health, induce weight loss, and increase insulin sensitivity and glycemic tolerance, as well as improve insulin resistance [238,239,240,241]. Reducing protein and amino acid content in the diet provides PD patients with an inexpensive and sustainable dietary intervention.

## 6. Perspectives

To sum up, we have reviewed the physiological benefits of DR, especially on PD, through various mechanisms including reducing PD-related risk factors, alleviating oxidative stress, improving mitochondrial function, maintaining autophagy homeostasis, and perfecting the gut microbiome, among others. The major concern of such a regimen is poor adherence in daily life and few guidelines have included DR for treating neurodegenerative diseases thus far. Further research is required to improve our understanding of the numerous DR protocols in PD and the underlying mechanisms. More clinical trials are necessary to assess whether DR could be an effective adjunctive therapeutic measure to the current medications.

## Figures and Tables

**Figure 1 nutrients-14-04108-f001:**
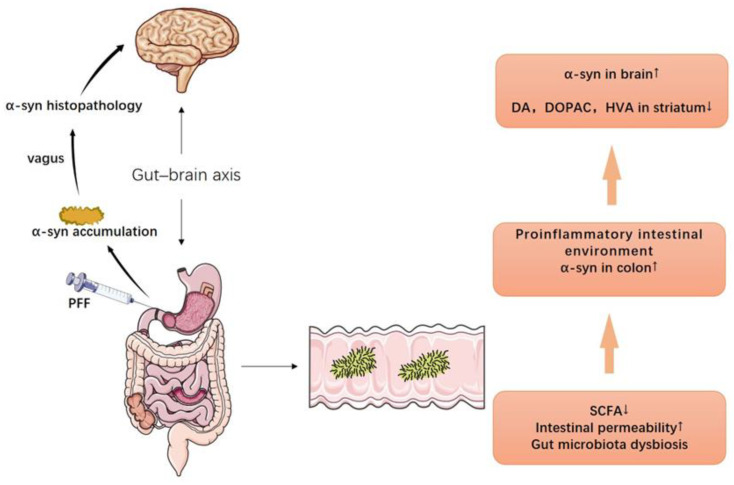
The gut–brain axis and the PD model of PFF. Injection of α-synuclein preformed fibers (PFFs) into duodenum and gastric pylorus induces α-synuclein aggregates’ formation in the dorsal motor nucleus of the vagus (DMV) and locus coeruleus, which eventually travel to the brain via the vagus nerve (the left half). Decreased SCFAs in gut, increased intestinal permeability, and intestinal microbiota imbalance cause a proinflammatory intestinal environment and α-synuclein aggregation, which ultimately lead to an increase in α-synuclein and a decrease in dopamine and dopaminergic metabolites (DOPAC, HVA) in the brain (the right half).

**Figure 2 nutrients-14-04108-f002:**
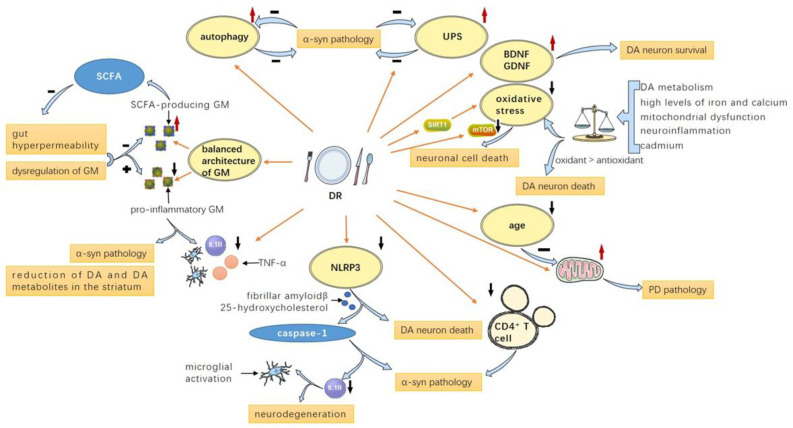
Possible mechanisms of DR on PD: DR contributes to a balanced architecture of GM consisting of more SCFA-producing GM and less pro-inflammatory GM, as the former leads to less gut hyperpermeability and the latter leads to increased inflammatory factor levels, α-synuclein pathology, and reduction in DA and DA metabolites in the striatum; DR attenuates NLRP3 inflammasome and CD4+ T cells, leading to reduced α-syn pathology and caspase-1 and IL-1β levels, which could promote microglial activation and neurodegeneration; DR also delays aging and enhances mitochondrial function, thus ameliorating PD pathology; DR suppresses oxidative stress by modulating mTOR and SIRT1-related pathways and boosts the level of neurotrophic factors like BDNF and GDNF; DR enhances autophagy and UPS to facilitate α-synuclein degradation, thus ameliorating α-synuclein pathology. PD: Parkinson’s disease; SCFA: short-chain fatty acid; DR: dietary restriction; GM: gut microbiota; UPS: ubiquitin-proteasome system; DA: dopamine; BDNF: brain-derived neurotrophic factor; GDNF: glial-cell-line-derived neurotrophic factor; α-syn: α-synuclein.

**Table 1 nutrients-14-04108-t001:** The physiological benefits of different types of DR.

	Regimen	Start	Duration	Physiological Benefits	Reference
Aging and Longevity					
Humans					
healthy humans	14% CR		2 years	↑transcriptional reprogramming in adipose tissue-regulating mitochondrial bioenergetics, anti-inflammatory responses, longevity ↓age-related inflammation	[27]
healthy, non-obese adults	ADF	35 to 65 years	4 weeks	↑cardiac health ↓body weight, fat-to-lean ratio, sICAM-1 (an age-associated inflammatory marker), low-density lipoprotein, metabolic regulator triiodothyronine	[42]
Animals					
Wistar strain male albino rats	ADF	21 months	3 months	↓age-associated impairment in motor coordination and learning and memory function, age-related increase in protein carbonylation, age-related impairment of synaptic proteins	[88]
rats	ADF	2 months	4 months/10 months/22 months	↓age-related oxidative damage, age-related increase in lipid peroxidation markers, age-related increase in TGF-β1 and collagen	[120]
C. elegans	ADF			↑lifespan	[121]
Drosophila melanogaster	TRF	2 weeks	5 weeks	↑sleep, age-induced decline in cardiac function	[43]
A/J mice	ADF	1.5 months		↑mean and maximum life span	[69]
male SD rats	60% CR	18 months	6 months	↑survival rate, spontaneous locomotor activity, spatial learning and reference memory, spatial cognition	[104]
Metabolism					
Humans					
humans with metabolic syndrome	69% CR 2 days a week	30–50 years	8 weeks	↑vasodilatory parameters, production of short-chain fatty acids (SCFAs) ↓oxidative stress, inflammatory cytokines, circulating levels of lipopolysaccharides (LPS)	[122]
humans with metabolic syndrome	TRF	18 years old or older	4 weeks	↑levels of tumor suppressor and DNA repair gene protein products (GP)s (CALU, INTS6, KIT, CROCC, PIGR), key regulatory proteins of insulin signaling (VPS8, POLRMT, IGFBP-5) ↓levels of tumor promoter GPs (POLK, CD109, CAMP, NIFK, SRGN), body mass index, waist circumference, blood pressure	[123]
males with obesity	TRF	20–30 years	30 days	↑leptin ↓body mass, BMI, body fat percentage (BFP), fat-free mass (FFM) and waist-to-hip ratio (WHR), glucagon-like peptide-1 (GLP-1), peptide YY (PYY), cholecystokinin (CCK)	[58]
humans	40% CR	mean age 53.0 ± 11 years	average of 6.9 ± 5.5 years	↑insulin sensitivity, serum adiponectin concentration, fructosamine, sRAGE, fasting serum free fatty acids ↓body fat and trunk fat, serum resistin concentration, serum IL-6, soluble TNF R-I and TNF R-II, plasma 2-h insulin and C-peptide concentrations	[26]
overweight and obese humans	ADF/75% CR	18 to 65 years	24 weeks	↑fat-free mass (FFM): total mass ratio ↓circulating leptin	[124]
humans with BMI 30–45	Intermittent/continuous CR	21–70 years	1 year	↑weight loss ↓cardiovascular risk factors, waist circumference	[125]
Animals					
mice	2:1 IF regimen (1 day of fasting followed by 2 days of feeding)	8 weeks	16 weeks	↑adipose thermogenesis contributing to IF-mediated metabolic benefits, adipose vascular endothelial growth factor expression ↓diet-induced metabolic abnormalities	[126]
mice	30% CR	10 weeks	≥5 weeks	↑endogenous fatty acid (FA) synthesis, FA oxidation, expression of FA synthase and acetyl-CoA carboxylase mRNA	[127]
C57BL/6N mice	ADF	7 to 8 weeks	15 weeks	↑energy expenditure, beiging of inguinal white adipose tissue ↓Ucp1 and Pgc1a mRNA	[128]
mice with obesity-induced diabetes	ADF	20 weeks	6 weeks	↑glucose-stimulated insulin secretion, beta cell survival, nuclear expression of NEUROG3 (a marker of pancreatic regeneration), autophagic flux in islets, glucose tolerance	[129]
C57BL/6 mice	40% DR	3 months	3/9/12 months	↓DNA damage, adipocyte size (area and perimeter) in visceral adipocytes	[70]
Nervous system					
Humans					
men and women	TRF	50 years or older	6 months	↓cognitive impairment	[130]
humans with central obesity	IER/CER	35–75 years	4 weeks	↑cognitive function, pattern separation	[106]
Animals					
rhesus monkeys with PD	30% CR	9–17 years	6 months	↑locomotor activity, dopamine (DA) and DA metabolites in the striatal region, glial-cell-line-derived neurotrophic factor	[105]
C57BL/6N mice	TRF/ADF	3 months	3 months	↑activation of the Notch signaling pathway (Notch 1, NICD1, and HES5), BDNF, cAMP response element-binding protein (p-CREB), expression of postsynaptic marker, PSD95, neuronal stem cell marker, Nestin, in the hippocampus	[131]
MPTP-induced PD mice	FMD	7 weeks	3 weeks	↑motor function, levels of BDNF ↓loss of dopaminergic neurons in the SN, the number of glial cells, the release of TNF-α and IL-1β	[132]
mice	ADF	2 months	3 months	↑BDNF in the hippocampus, striatum, and cerebral cortex	[112]
mice	ADF	7 weeks	11 months	↑drebrin and expression of synaptophysin in the cerebral cortex and hippocampus ↓blood cholesterol, triglycerides, high-density lipoproteins (HDL) and low-density lipoproteins (LDL) in the blood, glutathione disulfide (GSSG), 4-hydroxy-2-nonenal (HNE) and nitrotyrosine-containing proteins in the cerebral cortex	[118]
mice	ADF	2 months	3 months	↑neurogenesis, BDNF protein levels	[113]
mice	ADF	8 weeks	4 weeks	↑diversity of the gut microbiome, adiponectin levels, corticosterone levels, β- hydroxybutyrate ↓EAE clinical course and pathology, production of pro-inflammatory T cell cytokines, serum leptin	[133]
mice	TRF	10 weeks	3 months	↑cell proliferation in the intact subventricular zone (SVZ) ↓serum levels of leptin, sensorimotor impairment and infarct size after ischemia and reperfusion, stroke-induced cell proliferation in the hippocampus	[134]
mice	TRF		until 2 to 3 months	↑autophagy in the ventromedial nucleus of the hypothalamus	[135]
rat	50% DR every other day, and fed with vegetables on days in between	6 months	6/12/18 months	↓age-related a-synuclein expression	[136]
rat	ADF	3 months	3 months	↓levels of glucocorticoid receptor mRNA and protein in the hippocampus and cerebral cortex	[137]
PDAPP-J20 transgenic mice (AD model)	60%CR	6.5 months old	6 weeks	↑astroglial positive signal for LC3 ↓cognitive deficits, amyloid pathology and microglial reactivity	[56]
rat MS model	ADF	180–200 g	8 weeks	↑ TGF-β1 ↓IL-6, MMP-2 activity	[138]
mouse with traumatic brain injury	ADF	6–7 weeks	30 days	↑SIRT1 levels in the cortex and hippocampus ↓impaired hippocampus-dependent learning and memory	[139]

PD: Parkinson’s disease; CR: calorie reduction; ADF: alternate day fasting; TRF: time-restricted feeding; IER: intermittent energy restriction; CER: continuous energy restriction; FMD: fasting mimicking diet; AD: Alzheimer’s disease. The ↑ refers to the promoting effect, and the ↓ refers to the inhibiting effect.

## Data Availability

Not applicable.

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
