# Peer review of "Dietary Restriction against Parkinson’s Disease: What We Know So Far"

_nutrients, 2022, doi:10.3390/nu14194108_

Round 1
Reviewer 1 Report
The authors consider a vey important topic: nutrition in neurological diseas.
The paper is valuable and well written.
Please consider that PD affects older adults. For this reason I suggest to consider and the importance of nutrition on other aspects like as health and mortality (10.1093/gerona/glp183); depression (10.1016/j.jamda.2019.01.128; 10.1017/S1041610217001715)
The paper is suitable for publication in nutrients.
Reviewer 2 Report
Wang et al. prepared a review, where they discuss about the role of dietary intervention on the management of Parkinson’s disease (PD). Although the topic is interesting, the quality of the paper is not good enough for publication in the ‘nutrients’. The paper should be improved greatly before publication. My specific comments are given below:
1. Reading the paper, it feels like the authors wanted to say that pharmacological management of PD is not good enough (Of note, these pharmacologic strategies have been reported to cause early gastrointestinal and psychiatric adverse reactions, and vital drug withdrawal has also been observed in certain patients. While most pharmacological treatments fail to achieve satisfactory clinical outcomes in the neurodegenerative diseases, lifestyle interventions have risen in the spotlight) But dietary intervention might be better for the management of PD. Also, the authors said that the pharmacological treatment has some adverse effects. The reference the authors cited did not emphasize the adverse effects pharmacological treatment. Anyway, pharmacological treatments show some adverse effects, so dietary intervention might be good. Such statement is not acceptable in the medical scientific community. Dietary intervention may add some benefit, but not be the primary tool for the management of neurodegenerative diseases including PD. I suggest the authors to rewrite the paper focusing the dietary intervention of PD as an adjunct to pharmacological therapy. Of course, some adverse effects of pharmacological therapy can be seen, but that does not mean that we need to totally abandon the therapy and focus on lifestyle changes.
2. Parkinsonian diseases comprise several distinct diseases, where some of the diseases may not respond with conventional pharmacological PD therapy. But PD patients usually respond well with conventional PD therapy.
3. In the section 2 (Methods of DR), the authors need to rewrite the section by categorizing the types of dietary restriction. Then they can describe what type of physiological and molecular changes the particular DR used to do. Then they can say what types of diseases this particular pathway is important and can be modified by DR.
4. The heading of the section 3 is ‘Effects of DR on PD and the underlying mechanisms’. But in subsection 3.1.1, the effects of DR in life expectancy were described. This section should be moved to section 2, where how DR affects life expectancy should be discussed.
5. The authors can make a separate section using subsection 3.1.2 to 3.1.5. Also, the text should be more specific for the topic.
6. In the beginning of section 3, the pathology of PD should be described, with the underlying molecular mechanisms
7. Overall, the manuscript should be thoroughly rewritten and checked by a scientific editor.
Author Response
Response to Reviewer 2 Comments
Dear Editor and Reviewers,
Thank you for offering us an opportunity to improve the quality of our submitted manuscript (Dietary restriction against Parkinson’s disease: what we know so far). We appreciated very much for your constructive and insightful comments. In this revision, we have addressed all of these suggestions. We hope the revised manuscript has now met the publication standard of your journal. We highlighted all the revisions in yellow color.
Point 1: Reading the paper, it feels like the authors wanted to say that pharmacological management of PD is not good enough (Of note, these pharmacologic strategies have been reported to cause early gastrointestinal and psychiatric adverse reactions, and vital drug withdrawal has also been observed in certain patients. While most pharmacological treatments fail to achieve satisfactory clinical outcomes in the neurodegenerative diseases, lifestyle interventions have risen in the spotlight) But dietary intervention might be better for the management of PD. Also, the authors said that the pharmacological treatment has some adverse effects. The reference the authors cited did not emphasize the adverse effects pharmacological treatment. Anyway, pharmacological treatments show some adverse effects, so dietary intervention might be good. Such statement is not acceptable in the medical scientific community. Dietary intervention may add some benefit, but not be the primary tool for the management of neurodegenerative diseases including PD. I suggest the authors to rewrite the paper focusing the dietary intervention of PD as an adjunct to pharmacological therapy. Of course, some adverse effects of pharmacological therapy can be seen, but that does not mean that we need to totally abandon the therapy and focus on lifestyle changes.
Response 1: Please provide your response for Point 1. (in red)
Thank you very much for this valuable suggestion. We have reorganized the statement, presenting the current treatment strategies in a more objective way and adding more information of the adverse reactions of the classical medications. The modified content is in lines42-53.
Point 2: Parkinsonian diseases comprise several distinct diseases, where some of the diseases may not respond with conventional pharmacological PD therapy. But PD patients usually respond well with conventional PD therapy.
Response 2: Please provide your response for Point 2. (in red)
Thank you very much for your suggestion. We do understand that the current drugs are fairly effective in controlling the motor and non-motor symptoms of PD and PD patients respond well to the conventional medications. But those drugs can neither prevent PD onset nor stop the disease progression in the long term. In this study, we hope to present the current evidence of DR not only as an adjunct to pharmacological therapy, but also as a promising strategy to prevent the disease progression at the beginning.
Point 3: In the section 2 (Methods of DR), the authors need to rewrite the section by categorizing the types of dietary restriction. Then they can describe what type of physiological and molecular changes the particular DR used to do. Then they can say what types of diseases this particular pathway is important and can be modified by DR.
Response 3:
Thank you very much for this valuable suggestion. We have rewritten the section by categorizing the types of dietary restriction in details in lines 71-111.
Point 4: The heading of the section 3 is ‘Effects of DR on PD and the underlying mechanisms’. But in subsection 3.1.1, the effects of DR in life expectancy were described. This section should be moved to section 2, where how DR affects life expectancy should be discussed.
Response 4:
Thank you very much for this valuable suggestion. We have moved the part to the section 2, in lines 150-178 of the text.
Point 5: The authors can make a separate section using subsection 3.1.2 to 3.1.5. Also, the text should be more specific for the topic.
Response 5:
Thank you very much for this valuable suggestion. We have separated the section and keep the content specific for the topic. The modified content is in lines217-277.
Point 6: In the beginning of section 3, the pathology of PD should be described, with the underlying molecular mechanisms
Response 6:
Thank you very much for this valuable suggestion. We have described the pathology and the underlying mechanisms of PD that the beginning of the section. The added content is in lines280-285.
Point 7: Overall, the manuscript should be thoroughly rewritten and checked by a scientific editor.
Response 7:
Thank you very much for this valuable suggestion. The manuscript has been thoroughly rewritten and checked.

Reviewer 3 Report
In the present study, the authors summarized the current knowledge of the efficacy of dietary restriction (DR) on Parkinson’s disease (PD), focusing on the underlying mechanisms involving general metabolism, neuroendocrinolgy, neuroinflammation, gut microbiome, etc. We anticipate that this review will provide future perspectives for PD prevention and treatment.
This review is interesting and discusses an important topic for PD. Unfortunately, this manuscript needs substantial improvements and corrections before publishing may be possible.
General points:
Please add a list of abbreviations before References section to your manuscript.
Please correct all spaces between the words and references numbers according to your list of references in the text of your review.
Special points:
This manuscript should be substantially improved, i. e., by substantial references in the field.
Keywords: please add also to keywords: patients; treatment; prevention
Introduction
Line 31: Please say: Parkinson’s disease (PD) is the 2nd most common neurodegenerative disease…
Lines 31-43: please add more references at the end of each of these sentences.
Lines 38-45: please describe more exactly the up to date treatment of PD and include the appropriate references.
Lines 45-54: please describe exactly all these studies.
Main part of the manuscript:
Lines 57-58: please add multiple references at the end of this sentence.
Lines 57-78: please describe exactly all these studies.
Lines 80-81: please add multiple references at the end of this sentence.
Lines 80-83: please describe exactly all these studies.
Lines 96-103: please add more references at the end of each of these sentences.
Lines 123-130: please add multiple references at the end of each of these sentences.
Lines 138-141: please add multiple references at the end of each of these sentences.
Lines 162-166: please add multiple references at the end of each of these sentences.
Lines 187-188: please add multiple references at the end of this sentence.
Lines 207-218: please add multiple references at the end of each of these sentences.
Lines 230-248: please add more references at the end of each of these sentences.
Lines 256-257: please add multiple references at the end of this sentence.
Lines 256-278: please describe exactly all these studies.
Lines 280-281: please add multiple references at the end of this sentence.
Lines 280-309: please add and discuss another important risk factors of PD and a probable beneficial influence of DR on these factors.
Lines 312-315: please add multiple references at the end of this sentence.
Lines 350-352: please add multiple references at the end of this sentence.
Lines 355-362: please add multiple references at the end of each of these sentences.
Lines 396-398: please add multiple references at the end of each of these sentences.
Lines 423-434: please add multiple references at the end of each of these sentences.
Lines 457-461: please add multiple references at the end of this sentence.
Lines 461-470: please describe exactly all these studies.
Lines 472-490: please add multiple references at the end of each of these sentences.
Perspectives
Please add to your Perspectives section the recommendations for clinical practice and physicians.
Table A1: please add also publications with PD patients, Alzheimer’s disease patients, MS patients and another neurodegenerative disease.
Figure A1: please add the description of this figure to your Legend Figure A1.
Round 2
Reviewer 2 Report
I do not see any improvement of the manuscript. So, I have to reject the manuscript.
Reviewer 3 Report
Thank you for all corrections. They are fine.
One important point: once again, please correct all spaces between the words and references´ numbers.